

# A new species of the genus *Sarsamphiascus* Huys, 2009 (Copepoda: Harpacticoida: Miraciidae) from a sublittoral zone of Hawaii

Jisu Yeom and Wonchoel Lee

Department of Life Science, Hanyang University, Seoul, South Korea

## ABSTRACT

A new species of *Sarsamphiascus Huys, 2009* was collected from sandy sediments of Hawaii at 12 –18 m depth. While the new species, *Sarsamphiascus hawaiiensis* sp. nov., is morphologically most closely related to *S. kawamurai* (*Ueda & Nagai, 2005*), the two species can be distinguished by the combination of the following morphological characteristics: elongated segments of the antennule in the new species, type of outer setae of the P5 exopod (bare in *S. kawamurai*), position of the inner seta of the P5 exopod in both sexes (more proximal in *S. kawamurai*), length and type of the setae of female P6 (shorter and bare in *S. kawamurai*). This is the first species of *Sarsamphiascus* from Hawaii to be discovered. Molecular analyses of mitochondrial cytochrome c oxidase subunit I (mtCOI) and nuclear 18S ribosomal RNA (18S rRNA) genes confirmed that *S. hawaiiensis* and *S. kawamurai* are distinct species.

# INTRODUCTION

Miraciidae Dana, 1846 is a large family of benthic copepods in the order Harpacticoida (*Boxshall & Halsey, 2004*). It includes 50 valid genera and 426 species (*Song, Rho & Kim, 2007*; *Wells, 2007*; *Huys & Mu, 2008*; *Chullasorn et al., 2011*; *Karanovic & Cooper, 2012*; *Sönmez, Sak & Karaytuğ, 2014*), most of which were transferred from the former family Diosaccidae by *Willen (2002)*. The members of Miraciidae mostly inhabit the marine environment, although a few species occur in fresh water (*Boxshall & Halsey, 2004*). In Hawaii, two miraciid species, *Schizopera hawaiiensis* (*Kunz, 1995*) and *Macrosetella gracilis* (Dana, 1847), have been recorded (*Kunz, 1995*; *Eberl & Carpenter, 2007*).

The genus *Sarsamphiascus* was established by *Huys (2009)*, with *Dactylopus minutus* Claus, 1863 designated the type species. *Amphiascus* Sars, 1905 is a senior objective synonym of *Paramphiascopsis* Lang, 1944 and must be restricted to the species currently included in the latter. Therefore, the new genus *Sarsamphiascus* was proposed to receive all remaining "orphaned" species included in *Amphiascus* (*Huys, 2009*).

*Amphiascus* is a challenging genus in the sense that it is difficult to distinguish among species in this genus due to extreme inter- and intraspecific variation (*Lang, 1965*). *Lang (1948)* erected four groups (*minutus*, *varians*, *pacificus*, and *amblyops*) in *Amphiascus*

Corresponding author
Wonchoel Lee, wlee@hanyang.ac.kr

based on the swimming legs and A2 exopod setation patterns (*Hicks, 1989*). Species in the *pacificus*-group are characterized by having inner seta on P3 enp-2, no inner seta on P1 exp-2, and seta on A2 exp-1 (*Lang, 1948*). This group contains seven species: *S. parvus* (Sars G.O., 1906) from Europe, Asia and America; *S. kawamurai* (*Ueda & Nagai, 2005*) from East Asia and Turkey; *S. undosus* (*Lang, 1965*) from America; *S. pacificus* (Sars G.O., 1905) from Europe and New Zealand; *S. sinuatus* (Sars G.O., 1906) from Europe and America; and *S. humphriesi* (Roe, 1960) from Ireland. Each habitat and the distinguishing features of the *pacificus*-group and type species of the genus are shown in Table 1. We focused on the characteristics of female A1, A2, P5 and caudal terminal setae in this table. In the *pacificus*-group, the taxonomic validity of *S. humphriesi* (Roe, 1960) is uncertain because of the absence of inner seta on P2–P3 exp-1.

Today, there are 31 valid species and two inquirendum species in the genus worldwide (*Wells, 2007*; *Huys, 2009*). A new species of *Sarsamphiascus* was collected from SCUBA diving in Hawaii, U.S.A. Present study aims to describe the new species and to present an updated identification key to species of *Sarsamphiascus*.

## MATERIALS & METHODS

### Specimens collecting and observation

Specimens of the new species were collected from the Horseshoe Reef site of Oahu Island, Hawaii at 12–18 m depth on 6 August 2015 and 28 September 2018. Samples were collected by Wonchoel Lee.

Sediment samples were collected by SCUBA diving, sieved through a 38 µm sieve, and fixed with 99% ethanol. Harpacticoids were sorted under a dissecting microscope and stored in 99% ethanol. Harpacticoids were identified following *Huys et al. (1996)* and *Wells (2007)* using a compound microscope and 400–1,000×magnification. All drawings were prepared using a drawing tube on an Olympus BX51 differential interference contrast microscope.

The descriptive terminology of *Huys et al. (1996)* was adopted. Abbreviations used in the text are as follows: A1, antennule; A2, antenna; ae, aesthetasc; exp, exopod; enp, endopod; P1-P6, first to sixth thoracopod ; exp (enp)-1 (2, 3) to denote the proximal (middle, distal) segment of a three-segmented ramus. Specimens were deposited in the National Marine Biodiversity Institute of Korea (MABIK). Scale bars in figures are in µm.

We received *S. kawamurai* specimens from the MABIK marine zooplankton resource bank (REP000000000911) for comparative purposes. These specimens were collected from the West sea (Shin-An, SA/MRS002000107530) and the East sea (Go-Seong, GS/MRS002000107525; Yeong-Deok, YD/MRS002000107507; Gi-Jang, GJ/MRS002000107528) of South Korea and cultured by the Marine Zooplankton Resources Deposit Registration Preservation agency. Three of these specimens were randomly used to measure lengths of morphological features.

### Scanning electron microscopy (SEM)

In total, seven specimens were prepared for SEM. Materials were photographed using a Hitachi S-4700 scanning electron microscope at Eulji University and COXEM EM-30 at

**Table 1  Comparison of *pacificus*-group and the type species of the genus.**

| Species | Habitat | The number of P3 enp-2 inner setae | A2 exp-2 seta | P5 exp shape (length /breadth) | The length of P5 exp setae (3rd, 4th from inside) | P5 exp inner margin spinules | The length of P5 benp and inner-most seta | Caudal rami seta V inner margin | Caudal terminal setae type | A1 pinnate setae (location) |
|---|---|---|---|---|---|---|---|---|---|---|
| *S. minutus* | algae, net sweeping | 2 | O | oval (1.43) | <exp | O | >benp | straight | pinnate | O (seg-2) |
| *S. pacificus* | lagoon with brackish water | 1 | O | heart shape (1.38) | <exp | O | =benp | straight | naked | X |
| *S. sinuatus* | muddy sand sediment | 1 | O | oval (1.52) | mixed[a] | O | >benp | straight | pinnate | X |
| *S. parvus* | coral sand, subtidal zone sand | 1 | X | oval (1.38) | >exp | X | ≥benp | straight | pinnate | X |
| *S. undosus* | algae, fine sand | 1 | X | round (1.27) | >exp | O | <benp | undulating | pinnate | X |
| *S. kawamurai* | tidal pool, salt marshes, supralit-toral rocks | 1 | X | oval (1.65) | >exp | O | =benp | straight | naked | O (seg-1) |
| *S. humphriesi* | stone washing | 1 | O | oval (1.5) | mixed[a] | O | ≥benp | straight | – | X |
| *S. hawaiiensis* (New species) | sandy sed-iment | 1 | X | oval (1.7) | mixed[a] | O | <benp | straight | pinnate | O (seg-2) |

**Notes.**

*S. minutus* (Claus, 1863) is the type species of the genus (*minutus*-group); O: seta(e)/ spinule presence; X: seta(e)/ spinule absence.

[a]mixed: One seta is longer than exp, another is shorter.

Hanyang University. Digital photographs were processed and combined into plates using Adobe Photoshop CS6. To prepare specimens for SEM, they were transferred into pure isoamyl-acetate, critical-point dried, mounted on stubs, coated in gold, and observed under SEM on the in-lens detector at an accelerating voltage of 10.0 Kv and 15.0 Kv and working distance between 7.0 to 13.4 mm.

## DNA extraction and amplification

For DNA extraction and amplification, specimens were transferred into ultrapure water for two hours to remove ethanol. Specimens were then prepared for non-destructive DNA extraction in worm lysis buffer (*Williams et al., 1992*). Specimens were placed in tubes containing 25 μl lysis buffer and placed in a Takara thermocycler (Takara, Otsu, Shiga, Japan) with the following settings: 65 °C for 15 min, 95 °C for 20 min and 15 ° C for 2 min. After this, specimens were kept for morphological identification and describing them after genetically confirming. Unpurified total genomic DNA was kept at −20 ° C for long-term storage.

Fragments from two genes, the nuclear 18S ribosomal RNA (18S rRNA) and mitochondrial cytochrome oxidase subunit I (mtCOI) genes, were amplified using PCR premix (BiONEER Co.) and 3 μl of genomic DNA as template. PCR primers 18S-F1, 18S-F3, 18S-R7, and 18S-R9 (*Yamaguchi & Endo, 2003*) were used to amplify 18S ribosomal RNA. MtCOI was amplified with Cop-COI-2189 and LCO1490 primers (*Bucklin et al., 2010*). For 18S, the amplification protocol consisted of an initial denaturation at 94 °C for 5 min followed by 33 cycles of denaturation at 94 °C for 30 s, annealing at 47 °C for 30 s, and extension at 72 °C for 1 min; this was followed by a final extension step at 72 °C for 10 min. For mtCOI, the amplification protocol consisted of an initial denaturation at 94 °C for 5 min followed by 40 cycles of denaturation at 94 °C for 1 min, annealing at 45 °C for 2 min, and extension at 72 °C for 3 min; this was followed by a final extension step at 72 °C for 10 min (*Bucklin et al., 2010*). Successful amplification was confirmed by electrophoresis on a 1% agarose gel.

PCR products were sent to Macrogen (Seoul, Korea) for purification and DNA sequencing. DNA was sequenced on an ABI automatic capillary sequencer using the same sets of primers as used for amplification. All obtained sequences were visualized using Finch TV version 1.4.0 (https://digitalworldbiology.com/FinchTV; Geospiza Inc., USA). The quality of each sequence was evaluated and low resolution peaks were checked by comparing forward and reverse strands. BLAST searches revealed that the obtained sequences were copepod in origin and not contaminants. Sequence information from this study was deposited in the NCBI database (18S: MN496455, MN496456; MN541391–MN541394 and mtCOI: MN507530; MN542379, MN542380).

## Phylogenetic analyses

An additional 43 sequences were downloaded from GenBank and included in our analyses (Table 2). Obtained sequences were checked manually and aligned by the ClustalW algorithm (*Thompson, Higgins & Gibson, 1994*) in MEGA version 7.0 (*Kumar, Stecher & Tamura, 2016*). Phylogenetic analyses were performed using Neighbor-Joining (NJ),

**Table 2  GenBank numbers of sequences used in phylogenetic analyses in this study.**

| Gene marker | Species name | Accession no. | Reference |
|---|---|---|---|
| 18S | *Amonardia coreana* | KT030261 | SY Baek, UW Hwang, 2015, unpublished data |
| | *Amphiascoides atopus* | KC815328 | S Gomez et al., 2013, unpublished data |
| | *Diosaccus ezoensis* | KR048740 | SY Baek, UW Hwang, 2015, unpublished data |
| | *Miracia efferata* | EU380294 | *Huys, Mackenzie-Dodds & Llewellyn-Hughes (2009)* |
| | *Paramphiascella fulvofasciata* | EU380293 | *Huys, Mackenzie-Dodds & Llewellyn-Hughes (2009)* |
| | *Stenhelia* sp. | EU380291 | *Huys, Mackenzie-Dodds & Llewellyn-Hughes (2009)* |
| | *Typhlamphiascus typhlops* | EU380292 | *Huys, Mackenzie-Dodds & Llewellyn-Hughes (2009)* |
| | *Sarsamphiascus hawaiiensis* | MN496455 | This paper |
| | *Sarsamphiascus hawaiiensis* | MN496456 | This paper |
| | *Sarsamphiascus kawamurai* | MN541391 | This paper |
| | *Sarsamphiascus kawamurai* | MN541392 | This paper |
| | *Sarsamphiascus kawamurai* | MN541393 | This paper |
| | *Sarsamphiascus kawamurai* | MN541394 | This paper |
| | *Dactylopusia pauciarticulata* | KR048735 | SY Baek, UW Hwang, 2015, unpublished data |
| | Thalestridae sp. | MF077761 | *Khodami et al. (2017)* |
| COI | *Amonardia coreana* | KT030279 | SY Baek, UW Hwang, 2015, unpublished data |
| | *Amonardia normani* | MH242652 | M Leray & G Paulay, 2018, unpublished data |
| | *Amonardia perturbata* | MH242653 | M Leray & G Paulay, 2018, unpublished data |
| | *Amphiascoides atopus* | KF667526 | *Easton et al. (2014)* |
| | *Amphiascoides* sp. | MH242654 | M Leray & G Paulay, 2018, unpublished data |
| | *Amphiascopsis cinctus* | MH670487 | *Rossel & Arbizu (2019)* |
| | *Amphiascus* sp. | KX714910 | *Gollner et al. (2016)* |
| | *Bulbamphiascus imus* | MH670542 | *Rossel & Arbizu (2019)* |
| | *Delavalia palustris* | MH976534 | *Rossel & Arbizu (2019)* |
| | *Delavalia reflexa* | MH976545 | *Rossel & Arbizu (2019)* |
| | *Diosaccus ezoensis* | KR049013 | SY Baek, UW Hwang, 2015, unpublished data |
| | *Diosaccus spinatus* | MH242730 | M Leray & G Paulay, 2018, unpublished data |
| | *Eoschizopera* sp. | MH976580 | *Rossel & Arbizu (2019)* |
| | *Haloschizopera pygmaea* | MH976598 | *Rossel & Arbizu (2019)* |
| | *Haloschizopera* sp. | MH976605 | *Rossel & Arbizu (2019)* |
| | *Itostenhelia golikovi* | KF524864 | *Karanovic, Kim & Lee (2014)* |
| | *Itostenhelia polyhymnia* | KF524868 | *Karanovic, Kim & Lee (2014)* |
| | *Macrosetella gracilis* | MG742365 | P Santhanam, et al., 2017, unpublished data |
| | *Miracia efferata* | GU171350 | *Bucklin et al. (2010)* |
| | *Sarsamphiascus hawaiiensis* | MN507530 | This paper |
| | *Sarsamphiascus kawamurai* | MN542379 | This paper |
| | *Sarsamphiascus kawamurai* | MN542380 | This paper |
| | *Sarsamphiascus undosus* | MH242965 | M Leray & G Paulay, 2018, unpublished data |

**Table 2** (*continued*)

| Gene marker | Species name | Accession no. | Reference |
|---|---|---|---|
| | *Schizopera akation* | JQ390560 | *Karanovic & Cooper (2012)* |
| | *Schizopera akolos* | JQ390584 | *Karanovic & Cooper (2012)* |
| | *Schizopera analspinulosa* | JQ390588 | *Karanovic & Cooper (2012)* |
| | *Schizopera emphysema* | JQ390558 | *Karanovic & Cooper (2012)* |
| | *Schizopera knabeni* | KF667527 | *Easton et al. (2014)* |
| | *Schizopera kronosi* | JQ390567 | *Karanovic & Cooper (2012)* |
| | *Schizopera leptafurca* | JQ390590 | *Karanovic & Cooper (2012)* |
| | *Schizopera uranusi* | JQ390561 | *Karanovic & Cooper (2012)* |
| | *Stenhelia pubescens* | KF524870 | *Karanovic, Kim & Lee (2014)* |
| | *Stenhelia taiae* | KF524885 | *Karanovic, Kim & Lee (2014)* |
| | *Wellstenhelia calliope* | KF524872 | *Karanovic, Kim & Lee (2014)* |
| | *Wellstenhelia clio* | KF524873 | *Karanovic, Kim & Lee (2014)* |
| | *Wellstenhelia qingdaoensis* | KF524874 | *Karanovic, Kim & Lee (2014)* |
| | *Willenstenhelia thalia* | KF524882 | *Karanovic, Kim & Lee (2014)* |

Maximum Likelihood (ML) and Bayesian Inference (BI) approaches. NJ analysis used the Kimura two-parameter model (K2P) (*Kimura, 1980*; *Nei & Kumar, 2000*) with uniform rates. ML analysis used the K2+G+I model based on the model test result in MEGA. One thousand bootstrap replicates were performed to obtain a relative measure of node support for the resulting trees. A BI tree was constructed with MrBayes v3.2.6 x64 (Ronquist et al., 2012) based on the following model parameters obtained using jModelTest 2.1.10 (Darriba et al., 2012): nst = 6, rates = gamma, and ncat = 4. Markov Chain Monte Carlo (MCMC) was run with the following parameters: nchains = 4, ngen = 1,000,000, samplefreq = 100, and printfreq = 1,000. ML and BI trees were visualized using FigTree v1.4.2. Average pairwise distances were also computed in MEGA version 7.0 using the K2P model. All trees were rooted with thalestrid sequences.

The electronic version of this article in Portable Document Format (PDF) will represent a published work according to the International Commission on Zoological Nomenclature (ICZN), and hence the new names contained in the electronic version are effectively published under that Code from the electronic edition alone. This published work and the nomenclatural acts it contains have been registered in ZooBank, the online registration system for the ICZN. The ZooBank LSIDs (Life Science Identifiers) can be resolved and the associated information viewed through any standard web browser by appending the LSID to the prefix http://zoobank.org/. The LSID for this publication is: [urn:lsid:zoobank.org:pub:B93346F0-3942-4CC9-A6ED-D437D899239F]. The online version of this work is archived and available from the following digital repositories: PeerJ, PubMed Central, and CLOCKSS.

## RESULTS

Order Harpacticoida Sars G.O., 1903
Family Miraciidae Dana, 1846
Genus *Sarsamphiascus Huys, 2009*
*Sarsamphiascus hawaiiensis* sp. nov. (Figs. 1–6)
urn:lsid:zoobank.org:act:BACD08DD-8F16-4106-8FCE-F4C1ED96A8AC

Type locality. —Horseshoe Reef site, Oahu Island, Hawaii, U.S.A., 21°28′35.93″N, 158°13′30.57″W (depth: 14 m; sandy sediment; water temperature: 26–30 °C).

Material examined. —Holotype 1 ♀ (MABIKCR00246491) Allotype 1 ♂ (MABIKCR00246492). Paratype 2 ♀ ♀1♂ on three slides (MABIKCR00246493–MABIKCR00246495), 2 ♀ ♀ dissected on 14 slides (MABIKCR00246496, MABIKCR00246497), and 2 ♂ ♂ dissected on six slides (MABIKCR00246498, MABIKCR00246499). 3 ♀ ♀4♂ ♂ and 1 ♂ on two SEM stubs, respectively (MABIKCR00246500, MABIKCR00246501). Sampled by SCUBA diving on 6 Aug 2015 and 30 Sep 2018.

Description of female. —Total body length 544.3 µm ($n = 3$) (Figs. 1A, 1B); body slender, cylindrical, slightly tapering behind. Cephalosome bell-shaped. Rostrum (Figs. 1A, 1C). Large, tapering distally, about 2.8 times longer than broad, defined at base, reaching to distal margin of second antennulary segment; with two small sensilla. Anal somite (Fig. 1A) with row of spinules along distal margin; anal operculum well developed, semicircular, with fine setulae along its posterior margin.

Caudal ramus (Figs. 1A 1D, 6E, 6F). About 0.7–0.8 times as long as greatest width, armed with spinules inner distally and with short setules on distal margin (arrowed in Fig. 6F); each ramus armed with seven setae; seta I and seta II developed, located near distal corner of lateral margin; seta I spiniform; seta III bare, longer than seta II, located near distal corner of ventral surface; seta IV well developed, bipinnate, seta IV longer than half of seta V, seta V as long as urosome; both terminal caudal setae with fracture plane; seta VI bare, 1.5 times as long as seta II, slightly curved inward at its base; seta VII short, bare, located on dorsal surface of caudal ramus.

Antennule (Fig. 2C). Eight-segmented, slender; segment-1 with two spinules on inner edge; segment-2 about twice as long as broad, about 1.4 times as long as segment-1, and about 1.8 times as long as segment-3; segment-4 1.5 times as long as segment-3. Armature formula: 1-[1 bare], 2-[8 bare + 3 pinnate], 3-[7 bare], 4-[4 bare + ae], 5-[2 bare], 6-[4 bare], 7-[4 bare], 8-[7 bare + ae]. Aesthetasc on segment-4 fused basally with adjacent seta and about 2.3 times as long as distal four segments combined.

Antenna (Figs. 2D, 6B). Endopod two-segmented, enp-1 bearing pinnate seta, enp-2 with two spiniform setae and bare seta posteriorly, seven setae terminally, four geniculate setae, two bare setae and slightly unipinnate seta; exopod three-segmented, exp-1 about three times longer than broad, bearing pinnate seta distally, exp-2 very short without seta, exp-3 about 2.5 times longer than broad with seta proximally and three setae distally.

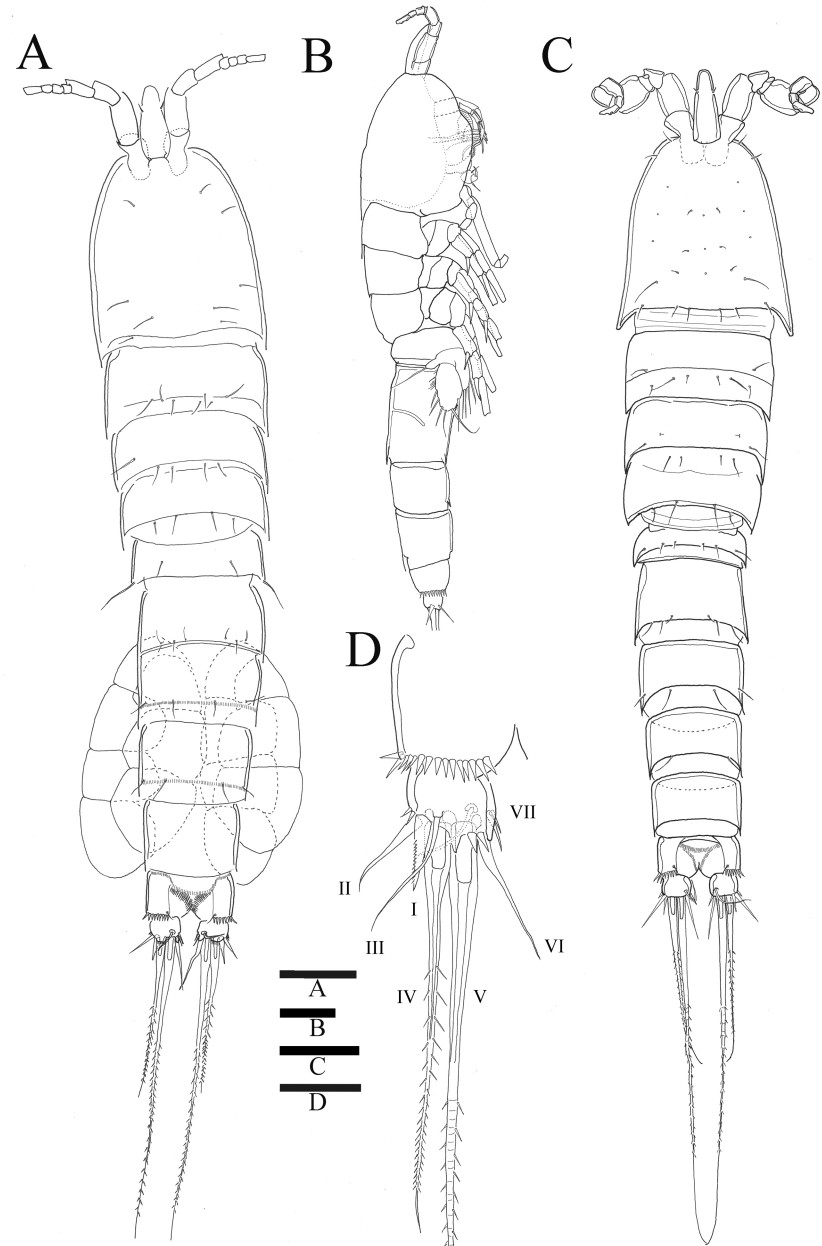

**Figure 1** *Sarsamphiascus hawaiiensis* **sp. nov.** (A) Habitus of female, dorsal. (B) Habitus of female, lateral. (C) Habitus of male, dorsal. (D) Caudal ramus of female, ventral. Scale bars: 50 μm (A –C), 20 μm (D).

Mandible (Fig. 2E). Gnathobase bearing chitinous projection on ventral surface; palp with three setae and setule rows; exopod two-segmented, exp-1 with pinnate seta and bare seta, exp-2 with two bare setae and pinnate seta; endopod one-segmented with two proximal and six distal setae.

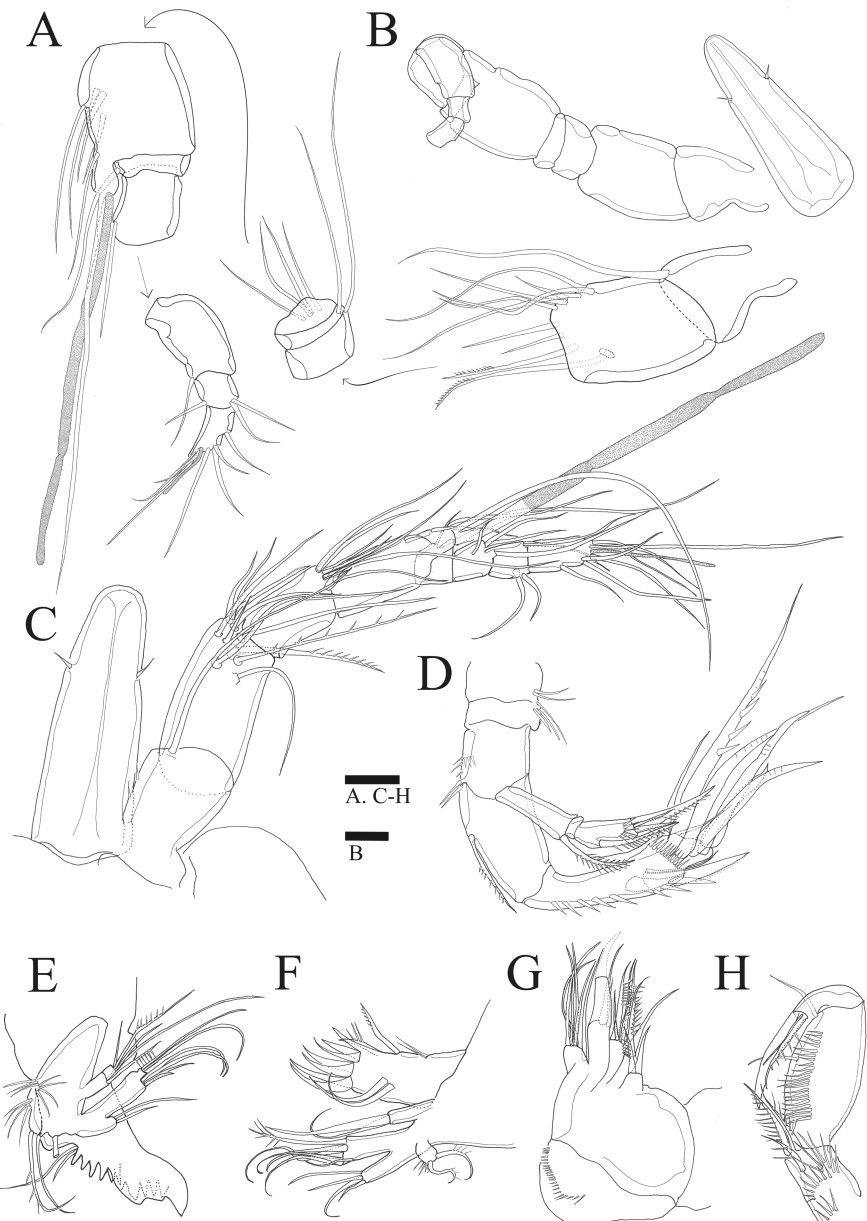

**Figure 2** ***Sarsamphiascus hawaiiensis* sp. nov.** (A–B), Male: (A) Segments of A1. (B) Antennule and Rostrum. (C–H) Female: (C) Antennule and Rostrum. (D) Antenna. (E) Mandible. (F) Maxillule. (G) Maxilla. (H) Maxilliped. Scale bars: 10 μm.

Maxillule (Fig. 2F). Praecoxal arthrite bearing seven elements distally with two pinnate setae and two bare setae on anterior surface; coxa with two setae; basis with six setae; endopod bearing four setae; exopod bearing two pinnate setae.

Maxilla (Fig. 2G). Three endites of syncoxa with four, two, two setae, respectively; basis with two thick setae and thin seta; endopod with six setae.

| | Endopod | Exopod |
|---|---|---|
| Table 3 | Setal formula of the new species. | |
| P1 | 1.0.030 | 0.0.023 |
| P2 | 1.2.121 | 1.1.123 |
| P3 | 1.1.321 | 1.1.123 |
| P4 | 1.1.221 | 1.1.323 |

Maxilliped (Fig. 2H). Subchelate; syncoxa with two long setae, relatively short seta and several rows of spinules; basis pectinate with several long spinules on the medial surface, twice as long as broad, bearing two bare setae; endopod elongate, with strong claw, bare seta, and unipinnate seta.

In P1–P4, all rami three-segmented and coxa ornamented with several rows of spinules. Setal formula of the new species is shown in Table 3.

P1 (Fig. 3A). Basis inner margin with setules and armed with bipinnate spine distally. Exp-1 inner margin bare, exp-1 and exp-2 with bipinnate outer spine and ornamented with spinules along outer margin, exp-2 and exp-3 inner margin with setules without inner seta, exp-3 with three outer spines and long geniculate seta and relatively short, geniculate and unipinnate setae distally. Enp-1 longer than exopod, about six times longer than wide with inner seta distally, inner and outer margin ornamented with setules; enp-2 small, ornamented with spinules along distal margin, without inner seta; enp-3 longer than enp-2, outer margin ornamented with spinules, bearing slender seta at inner distal corner, unipinnate claw-like spine and relatively short geniculate seta apically.

P2 (Fig. 3B). Basis with bipinnate outer spine. Exp-1 with seta at distal inner margin, setules on inner distal margin, exp-2 with inner seta at distal, inner margin ornamented with setules, both exp-1 and exp-2 with bipinnate outer spine and outer margin ornamented with spinules, exp-3 with plumose inner seta, long plumose seta at inner terminal, long seta with plumose inner side and pinnate outer side at outer terminal and three outer spines, proximal inner margin ornamented with setules, proximal outer margin ornamented with spinules. Endopod about as long as exopod, enp-1 with plumose inner seta, outer margin ornamented with setules; enp-2 with two inner setae; enp-3 with plumose inner seta, two setae distally and spine at outer distal corner, both enp-2 and enp-3 outer margin ornamented with spinules.

P3 (Fig. 4A). Basis with bare outer seta. Exp-1 with plumose inner seta, proximal inner margin ornamented with setules, exp-2 with plumose inner seta at distal margin, both exp-1 and exp-2 setules on inner distal margin and outer margin ornamented with spinules; exp-3 with plumose inner seta, long plumose seta at inner terminal, long seta with plumose inner side and pinnate outer side at outer terminal and three outer pinnate spines, proximal outer margin ornamented with spinules. Endopod about as long as exopod, both enp-1 and enp-2 with plumose inner seta and setules on inner distal margin, enp-3 with two plumose and unipinnate seta on inner side, two setae distally and spine at outer distal corner, outer margin of endopod segments ornamented with spinules.

P4 (Fig. 4B). Basis with bare outer seta. Both exp-1 and exp-2 with plumose inner seta, inner margin ornamented with setules, setules on inner distal margin and outer margin

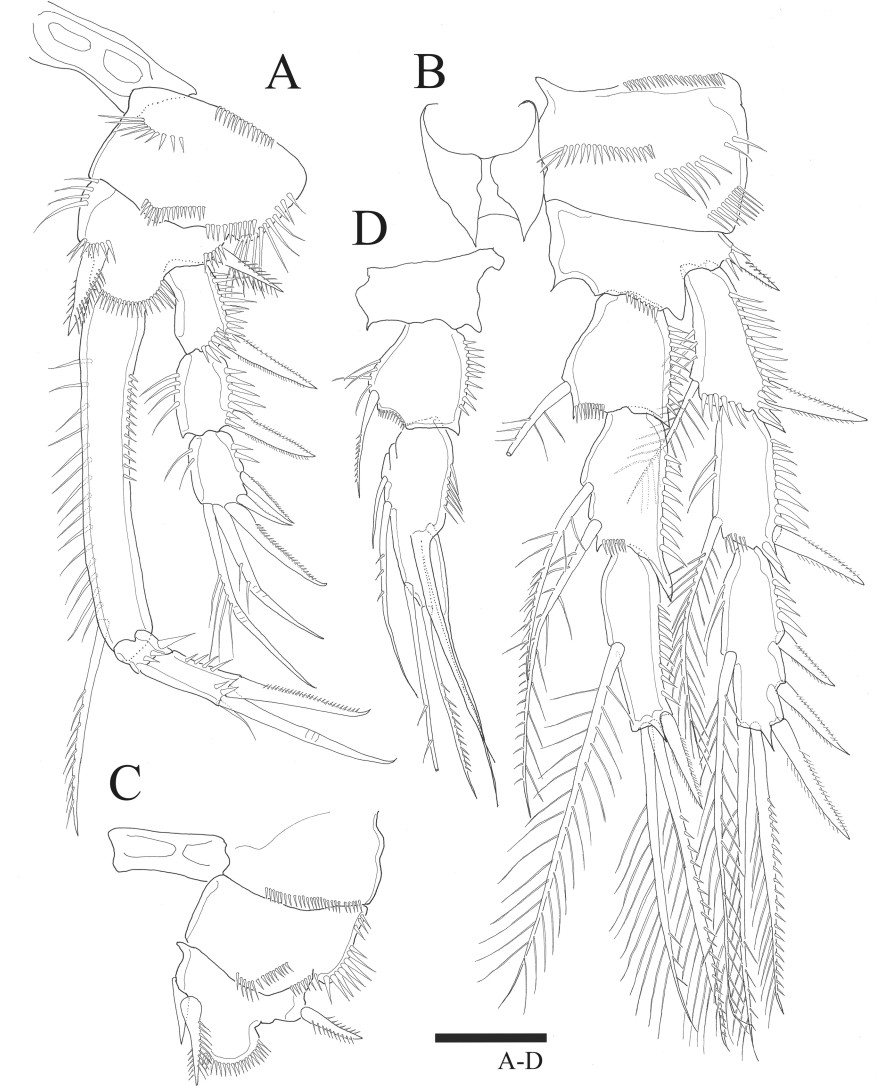

**Figure 3** *Sarsamphiascus hawaiiensis* **sp. nov.** (A –B) Female: (A) P1. (B) P2. (C –D) Male: (C) P2 endopod. (D) P1 basis. Scale bar: 20 μm.

ornamented with spinules; exp-3 with three plumose inner setae, the second seta of three inner setae thicker than others, long plumose seta at inner terminal, long seta with plumose inner side and pinnate outer side at outer terminal and three outer spines, proximal outer margin ornamented with spinules. Endopod longer than exp-1 and exp-2 combined, both enp-1 and enp-2 with plumose inner seta and setules on inner distal margin, enp-3 with two plumose inner setae, two setae distally and spine at outer distal corner, outer margin of endopod segments ornamented with spinules.

P5 (Figs. 5A, 6A). Benp and exopod distinct, benp with long, slender and bare outer basal seta; endopodal lobe bearing three inner bipinnate spines and two bipinnate distal setae. Exopod oval, elongated, 1.7 times longer than wide, with spinules along both inner and

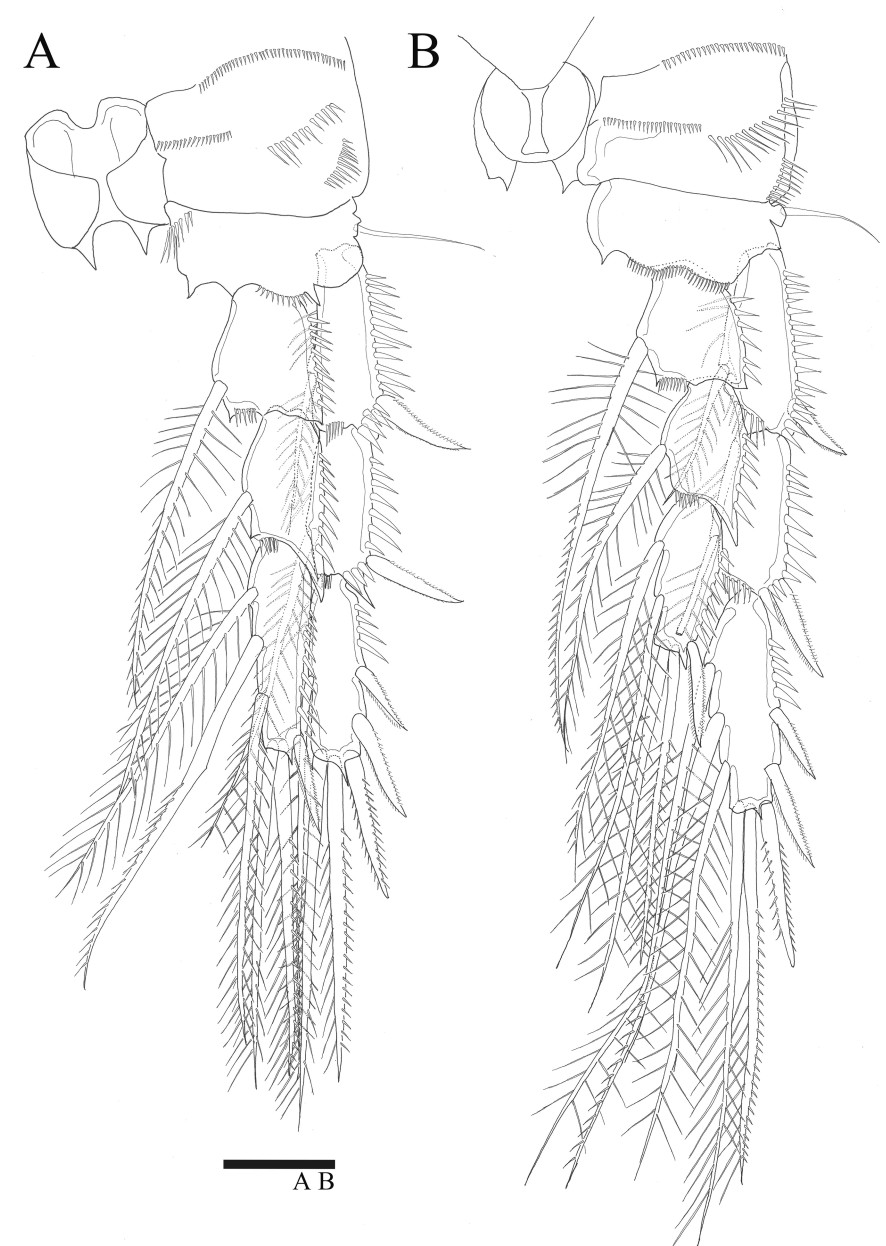

**Figure 4** *Sarsamphiascus hawaiiensis* **sp. nov.** Female: (A) P3. (B) P4. Scale bar: 20 μm.

outer margins. Exopod bearing bipinnate inner seta, two apical bare setae, three bipinnate outer setae and short bare seta.

Genital area as in Fig. 5B. P6 with long bipinnate seta, unipinnate seta, and short spiniform seta; kidney-shaped seminal receptacle (arrowed in Fig. 5B). Genital double-somite (Fig. 6A) with rows of spinules on ventrolateral surface.

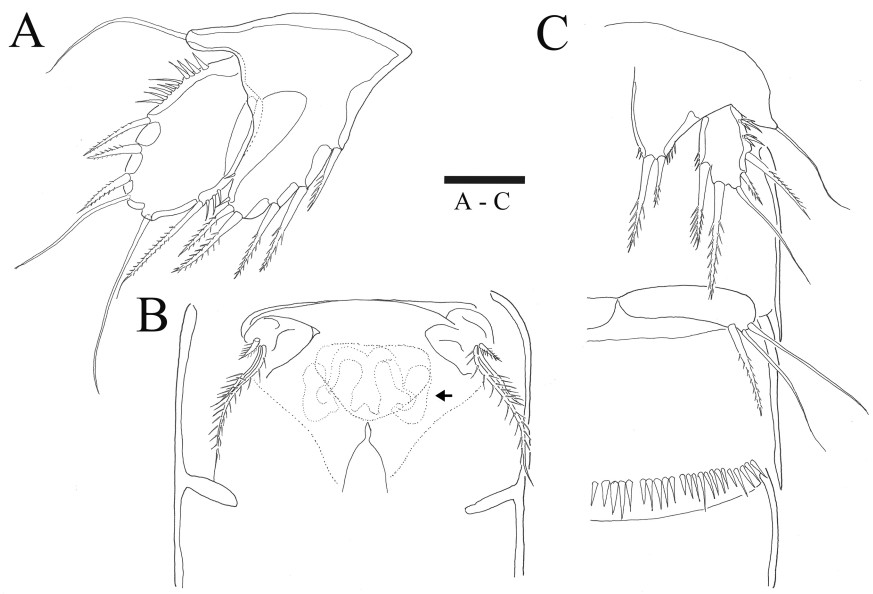

**Figure 5** *Sarsamphiascus hawaiiensis* **sp. nov.** (A–B) Female: (A) P5. (B) Genital field. C, Male: (C) P5, P6, and urosome (right), ventral. Scale bar: 20 µm.

Description of male. —Body (Fig. 1C). About 484.8 µm ($n = 3$) in length, smaller and slenderer than female. First and second abdominal somites each with spinule row near distoventral margins. Sexual dimorphism seen in A1, P1, P2, P5, and P6.

Antennule (Figs. 2A, 2B). Subchirocer, 10-segmented; armature formula: 1-[1 bare], 2-[8 bare + 2 pinnate], 3-[1 bare], 4-[5 bare], 5-[6 bare + ae], 6-[1 bare], 7-[1 bare], 8-[1 bare], 9-[3 bare], 10-[7 bare + ae].

P1 basis (Fig. 3C). With inner spiniform projection.

P2 endopod (Fig. 3D). Two-segmented; enp-1 with inner pinnate seta; enp-2 modified, bearing three inner setae, unipinnate seta distally, and two spiniform setae at mid-length of outer margin.

P5 (Figs. 5C, 6C, 6D). Benp and exopod distinct; endopodal lobe ornamented with spinules at inner and outer margin, armatured with two bipinnate spines. Inner spine twice as long as outer one. Exopod with five setae in total, including two inner setae, bare long distal seta and two pinnate outer setae, outer margin ornamented with rows of three spines.

P6 (Fig. 5C). Represented by pinnate inner seta and two bare setae on outer distal corner of genital operculum.

Etymology. —The species name refers the type locality of the new species, namely Hawaii.

Molecular results. —DNA was extracted and the 18S and mtCOI fragments were successfully PCR-amplified from two specimens and one specimen of the new species, respectively (18S; 1647 bp, 1692 bp, mtCOI; 678 bp). In addition, 18S and mtCOI fragments were PCR-amplified from four specimens and two specimens of *S. kawamurai*, respectively.

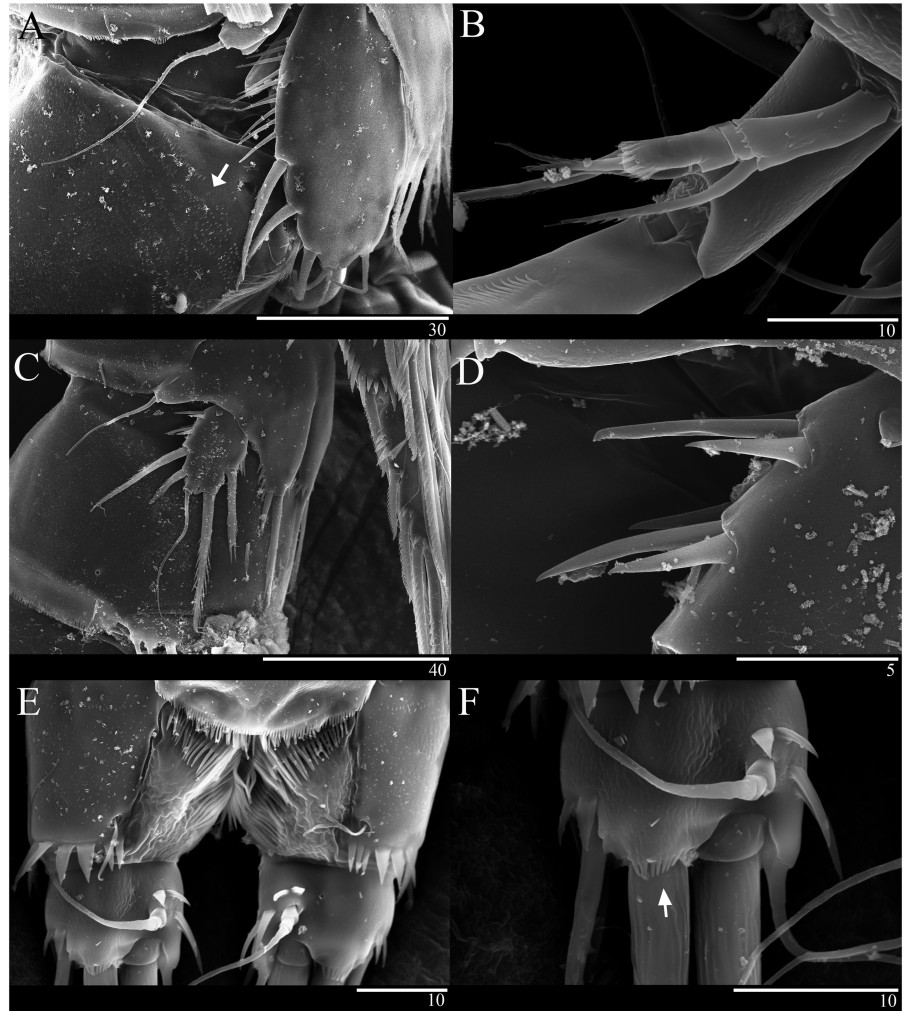

**Figure 6** ***Sarsamphiascus hawaiiensis* sp. nov., SEM photographs.** Female: (A) Genital double-somite and P5, lateral. Male: (B) A2 exp. (C) P5. (D) Rows of 3 spines on outer margin of P5. (E) Anal somite and Caudal rami, dorsal. (F) Caudal ramus, dorsal.

BLAST analyses of GenBank revealed that the obtained sequences were harpacticoid in origin and not contaminants. All analyses were run with all additional seven miraciid 18S sequences and 34 mtCOI sequences downloaded from GenBank (Table 2).

For the 18S analysis, sequences from species in eight different genera were obtained from the NCBI database: *Amonardia* Lang, 1944, *Amphiascoides* Nicholls, 1941, *Diosaccus* Boeck, 1873, *Miracia* Dana, 1846, *Paramphiascella* Lang, 1944, *Stenhelia* Boeck, 1865, *Typhlamphiascus* Lang, 1944, and *Sarsamphiascus*. Two thalestrid-morpha species, Thalestridae sp., *Dactylopusia pauciarticulata*, were selected as outgroups for 18S analysis. Average pairwise distance between the 13 miraciid sequences was 0.054. The pairwise distance between *Sarsamphiascus* and *Paramphiascella* 18S sequences was 0.022. This was the lowest value among the mean distances between *Sarsamphiascus* and the other seven genera (Table S1). Mean distance within *Sarsamphiascus* was 0.002. While the neighbor

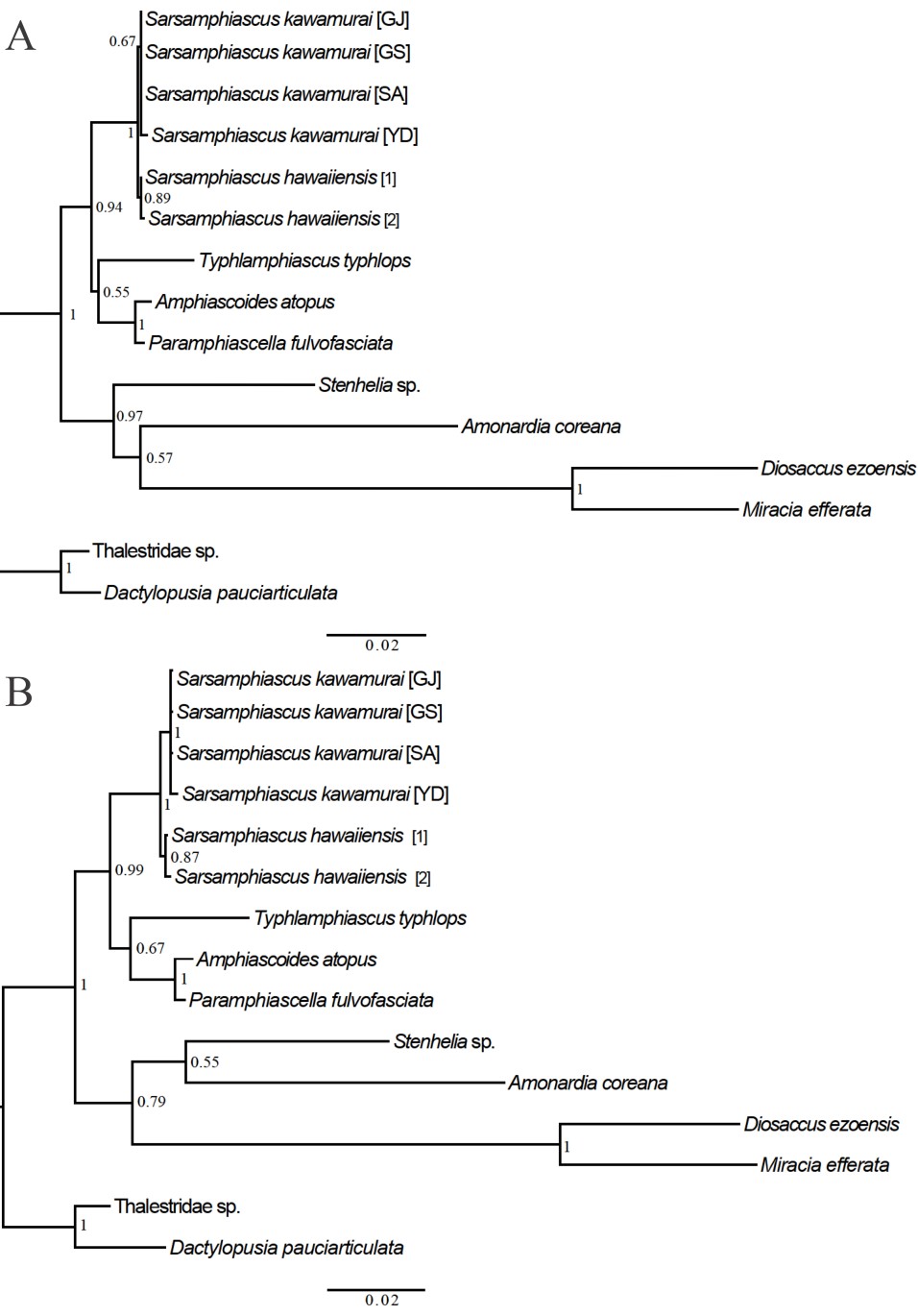

**Figure 7** **Phylogenetic trees of Miraciidae based on nuclear 18S ribosomal RNA data.** (A) Maximum Likelihood [ML] phylogenetic tree. (B) Bayesian Inference [BI] phylogenetic tree.

joining tree (Fig. S1) and maximum likelihood tree (Fig. 7A) had the same topology, the Bayesian inference tree (Fig. 7B) topology was different within *Amonardia* and *Stenhelia*.

In the mtCOI analysis, within *Sarsamphiascus*, the pairwise distance between the new species and *S. undosus* was 23.9%, and between the new species and *S. kawamurai* (two

**Table 4  Estimates of average evolutionary divergence over sequence pairs within Miraciidae genera based on mitochondrial cytochrome oxidase I (mtCOI) gene data.** This table only includes genera in which more than two species of data exist.

|  | Mean distance within genus | Standard error | Number of species included |
|---|---|---|---|
| *Amonardia* | 0.235 | 0.017 | 3 |
| *Amphiascoides* | 0.238 | 0.022 | 2 |
| *Delavalia* | 0.340 | 0.027 | 2 |
| *Diosaccus* | 0.232 | 0.021 | 2 |
| *Haloshizopera* | 0.256 | 0.020 | 2 |
| *Itostenhelia* | 0.055 | 0.010 | 2 |
| *Sarsamphiascus* | 0.206 | 0.014 | 4 |
| *Schizopera* | 0.244 | 0.013 | 8 |
| *Stenhelia* | 0.225 | 0.020 | 2 |
| *Wellstenhelia* | 0.231 | 0.018 | 3 |

specimens) was 24.1% and 25.8%, respectively (Table S2). These divergence values are within the range of mean distances within other miraciid genera (5.5–34%; overall mean distance: 31.5%) (Table 4).

## DISCUSSION

The new species described here clearly belongs to the genus *Sarsamphiascus* based on the presence of several diagnostic characters for the genus including P1 enp-1 longer than exp, P1 exp-3 with five setae, every exp-1, 2 with inner seta, P2 enp-2 with two inner setae, P3 enp-3 with three inner setae, and P4 enp shorter than exp. Within this genus, the *pacificus*-group has the following common characters: A2 exp-1 with seta and P1 exp-2 without inner seta.

The type species of this genus is *S. minutus* (Claus, 1863), originally reported as *Dactylopus minutus* from Helgoland, Germany. Conspicuous morphological differences between the type species and *S. hawaiiensis* sp. nov. are the presence of seta on A2 exp-2 and P1 exp-2, the length of P1 exp-2 and enp-3 segments, and the number of seta on P3 enp-2. A comparison of other morphological features is provided in Table 1.

Based on previous descriptions and keys to harpacticoid species (*Lang, 1948*; *Wells, 2007*), *S. hawaiiensis* is superficially similar to *Sarsamphiascus kawamurai* (*Ueda & Nagai, 2005*). This latter species was reported from outdoor laver cultivation tanks on the shore of Ariake Bay, Kyushu, Japan by *Ueda & Nagai (2005)*; tidal pool and salt marshes in the East Sea, Korea by *Chang (2009)*; and sublittoral rocks in the Aegean Sea and Mediterranean Sea of Turkey by *Sönmez, Sak & Karaytuğ (2014)*. According to the descriptions in the three reports mentioned above, the distinguishable characters of *S. kawamurai* are the absence of a ventral row of spinules on the genital double-somite, bare caudal terminal setae, pinnate seta on the first segment of A1, and short outer seta of the male P5 baseoendopod. *S. hawaiiensis* and *S. kawamurai* share the characters of setal formula, elongated P5 exp, and presence of an inner margin with spinules on the P5 exp. However, the new species and *S. kawamurai* can be distinguished by the combination of the following morphological

characteristics: (1) elongated segments of the antennule in the new species, especially the second and third segments of the antennule (twice as long as broad) (1.4 and 1.3 times in *S. kawamurai*, respectively); (2) type of outer setae of the P5 exopod (bare in *S. kawamurai*); (3) position of the inner seta of the P5 exopod in both sexes (more proximal in *S. kawamurai*); (4) length and type of the setae of female P6 (shorter and bare in *S. kawamurai*). This new species is the first report of the genus *Sarsamphiascus* in the Hawaiian Islands.

The new species and *S. undosus* share the characters of setal formula, short innermost seta of female P5 benp, and caudal rami terminal setae with ornamentation. However, the new species and *S. undosus* can be distinguished from each other by the combination of the following morphological characteristics: (1) ratio of length and width of female P5 exp (round in *S. undosus*); (2) length of outer setae of female P5 exp (longer in *S. undosus*); (3) type of two inner setae onP6 (bare in *S. undosus*); (4) seta type of the A1 s segment (bare in *S. undosus*); (5) type of inner edge of caudalterminal seta (undulate in the proximal part in *S. undosus*).

When these three species with similar morphologies were compared within the *pacificus*-group, ratios of the length to width of the rostrum, A1 segments, P5 exp, and caudal rami of the new species were found to be higher than those of the other two congeners: the ratio of rostrum length to width is 2.8 (vs. 2 in *S. kawamurai* and *S. undosus*); the ratio of the length to width of the second and third antennular segment is 2 (1.3–1.4 in *S. kawamurai* and *S. undosus*); the ratio of the length to width of the female P5 exp is 1.7 (1.65 in *S. kawamurai* and 1.3 in *S. undosus*); and the ratio of the length to width of the caudal rami is 0.75 (0.6 in *S. kawamurai* and 0.64 in *S. undosus*).

In addition, ornamentations of setae vary between these species. The new species has more ornamented setae on the second antennular segment, maxilliped, and P6. The new species also has ornamented setae on the second antennular segment and distal part of maxilliped. The two setae of P6 are pinnate (all bare in *S. kawamurai* and *S. undosus*).

All these characters suggest that the new species might have diverged early in the evolution of the *pacificus*-group compared with the congener species (*S. kawamurai* and *S. undosus*).

Comparison of the four species groups within *Sarsamphiascus* revealed that there are two inner setae on P3 enp-2 in the *minutus*-group versus seta in the other groups. The *minutus*–group and *varians*-group have inner seta on P1 exp-2 (absent in *pacificus*-group and *amblyops*-group). *Sarsamphiascus amblyops* (Sars G.O., 1911) in the *amblyops*-group exceptively has two setae on A2 exp-1 (seta in other groups). These reductions in swimming leg setation and A2 exp suggest that the *minutus*-group is the most primitive group within the genus followed by the *varians*-group. The *pacificus*-group is the most advanced group based on these characters. To evaluate if these morphological characters are homologous, more molecular data for the four species groups within this genus are required.

Analysis of 18S and mtCOI sequences revealed that the genera most closely related to *Sarsamphiascus* are *Paramphiascella* (0.022; Table S1) and *Eoschizopera* (0.257; Table S3). Since the types of molecular marker registered in NCBI differ by species, more DNA barcodes studies are necessary to establish a database for accurate comparison.

Since the genus *Amphiascus* has undergone many systematic changes, there are still several lineages that need to be studied using molecular data. The sequence information generated in this study is likely to be useful in future studies.

Three different methods were used to obtain phylogenetic trees based on the 18S data. While the neighbor-joining tree (Fig. S1) and maximum likelihood tree (Fig. 7A) had the same topology, the Bayesian inference tree (Fig. 7B) was different. In the Bayesian inference tree, *Amonardia* was more closely related to *Stenhelia* sp. than in the other trees. It is difficult to determine which result is more reasonable because bootstrap values for the clade containing *Amonardia* did not exceed 0.6 in any of the trees. However, support for the grouping of *Amonardia* with *Miracia* and *Diosaccus* was slightly higher as the NJ tree and ML tree. Nevertheless, common findings in all three trees were that all *Sarsamphiascus* species formed a monophyletic group and that *Sarsamphiascus* formed a clade with other Diosaccinae species with high probability. Furthermore, *Diosaccus* and *Amonardia* formed a clade with *Miracia* and *Stenhelia*, separated from subfamily Diosaccinae. The systematic positions of the three subfamilies relative to each other need to be redefined through analyses that include more miraciid sequences.

Pairwise distance results for the mtCOI gene support the distinctness of *S. hawaiiensis* sp. nov. from *S. undosus* and *S. kawamurai* and support that these species belong to the same genus based on the range of pairwise distance values seen in other genera within the family Miraciidae (Table 4).

Key to species of the genus *Sarsamphiascus* (modified from *Lang, 1948*; *Wells, 2007*)

We developed an updated key based on selected characteristics from the original description that identify species within the genus *Sarsamphiascus*.

1. P3 enp-2 with inner seta … 2
- P3 enp-2 with two inner setae … 4 <*Minutus* Group>
2. P1 exp-2 with inner seta … 15 <*Varians* Group>
- P1 exp-2 without inner seta … 3
3. A2 exp-1 with seta … 26 <*Pacificus* Group>
- A2 exp-1 with two setae … *S. amblyops* (Sars G.O., 1911) <*Amblyops* Group>
4. P2 exp-3 with two outer spines … *S. demersus* (Nicholls, 1939)
- P2 exp-3 with three outer spines … 5
5. Exp-3 of P3-P4 with eight setae; A2 exp-2 with seta … 6
- These characters not combined. … 7
6. Ratio of the length of P1 enp-3 to enp-2 is 3 … *S. longiarticulatus* (Marcus, 1974)
- Ratio of length of P1 enp-3 to enp-2 is 1.5 … *S. paracaudaespinosus* (Roe, 1958)
7. P3 enp-2 with two inner setae; A2 exp-2 without seta or A2 exp two-segmented … 8 - These characters not combined. … 9

8. A2 exp two or three-segmented; P3 exp-3 with seven setae; P4 exp-3 with eight setae; rostrum triangular and apex extremely finely pointed; P1 enp-3 length at least twice as long as enp-2 length …*S. ultimus* (Monard, 1928)

- A2 exp two or three-segmented; P3 exp-3 with seven setae; P4 exp-3 with eight setae; female, caudal rami longer than breadth with a dorsal ridge/ male, caudal rami much broader than length; P5 exp with five setae; length of P1 enp-2 and enp-3 approximately equal …*S. discrepans* (Mielke, 1989)

- A2 exp three-segmented; P3 exp-3 with eight setae (if seven setae, caudal rami approximately as broad as long and length of P1 enp-2 and enp-3 approximately equal) …*S. caudaespinosus* (Brian, 1927)

- These characters not combined. …9

9. P1 exp-2 normal, not extended …10

- P1 exp-2 extended …13

10. P5 exp of female with seven setae …*S. brevis* (Sars, 1909)

- P5 exp of female with seven setae …11

11. Ratio of P1 enp-1 length to breadth is 5; P5 benp, outer distal corner round in both sexes; P5 exp of male with six setae …*S. congener* (Sars, 1909)

- Ratio of P1 enp-1 length to breadth is 6 or more …12

12. Female, outer distal corner of P5 benp round / male, outer distal corner of P5 benp square …*S. graciloides* (Klie, 1950)

- Outer distal corner of P5 benp square in both sexes …*S. tenuiremis* (Brady, 1880)

13. P5 benp of female with four well-developed setae and short seta. …14

- P5 benp of female with five well-developed setae …*S. minutus* (Claus, 1863)

14. P5 exp of female with tapering distal edge …*S. hirtus* (Gurney, 1927)

- P5 exp of female round …*S. gracilis* (Lang, 1936)

15. P4 exp-3 with seven setae …*S. ampullifer* (Humes, 1953)

- P4 exp-3 with eight setae …16

16. P5 exp of female with five setae …*S. varians* (Norman & Scott 1905)

- P5 exp of female with six setae …17

17. P5 benp of female, seta I and II widely apart …18

- P5 benp of female, seta I and II not widely apart …19

18. First segment of A1 with spur …*S. dentiformis* (Coull, 1971)

- First segment of A1 without spur …*S. gauthieri* (Monard, 1936)

19. P5 benp of female does not extend beyond half the length of the P5 exp …20

- P5 benp of female extends beyond half the length of the P5 exp …22

20. P5 benp of female does not reach half the length of the P5 exp …*S. angustipes* (Gurney, 1927)

- P5 benp of female reaches half the length of the P5 exp …21

21. P5 exp of female at least twice as long as broad …*S. propinquus* (Sars, 1906)

- P5 exp of female not twice as long as broad …*S. polaris* (Sars, 1909)

22. P5 benp of female reaches to the end of the P5 exp …*S. elongatus* (Ito, 1972)

- P5 benp of female does not reach the end of the P5 exp …23

23. P1 enp-3 outer margin with spinules …24

- P1 enp-3 outer margin without spinules …*S. profundus* (Becker & Schriever, 1979)

24. P5 benp of male with three setae and outer distal corner of benp with an obvious spiniform seta …*S. tenellus* (Sars, 1906)

- P5 benp of male with two setae and outer distal corner of benp with a mucroniform projection …25

25. P1 enp-2 and enp-3 with bare seta …*S. tainui* (Hicks, 1989)

- P1 enp-2 and enp-3 with pinnate seta …*S. lobatus* (Hicks, 1971)

26. P2-P3 exp-1 with inner seta …27

- P2-P3 exp-1 without inner seta …*S. humphriesi* (Roe, 1960)

27. A2 exp-2 with seta; P5 benp without hyaline fields …28

- A2 exp-2 without seta …29

28. A2 exp-3 with inner seta; P2–P4 enp extending to at least the end of the exp; P5 exp of female oval …*S. sinuatus* (Sars, 1906)

- A2 exp-3 without inner seta; P2–P4 enp does not extend to at least the end of the exp; P5 exp of female tapers …*S. pacificus* (Sars, 1905)

29. P5 exp of female round; inner terminal seta of caudal rami undulating in proximal part of inner edge …*S. undosus* (*Lang, 1965*)

- P5 exp of female oval; inner terminal seta of caudal rami straight …30

30. P5 exp of female, both third and fourth setae from inside long and extend well beyond exp …31

-P5 exp of female, third seta from inside extend well beyond exp; fourth seta extends only to about the end of exp …*S. hawaiiensis* sp. nov.

31. A1 with pinnate seta on first segment; male P5 benp outer seta shorter than half the length of the inner seta …*S. kawamurai* (*Ueda & Nagai, 2005*)

- A1 without pinnate seta, all bare setae; male P5 benp outer seta as long as inner seta …*S. parvus* (Sars, 1906)

## CONCLUSIONS

A new species, *S. hawaiiensis* sp. nov., was collected from subtidal sandy sediments of Hawaii. The new species can be distinguished from its congeners in the *pacificus*-group by morphological characteristics and molecular analyses of mtCOI genes and 18S rRNA genes. This is the first species of *Sarsamphiascus* to be discovered from Hawaii.

In this genus, molecular data for only one of four species groups (the *pacificus*-group) is currently available in NCBI. To evaluate whether genus designations based on morphological characters are correct and to determine phylogenetic relationships among genera, more molecular data for the four groups need to be generated and analyzed.

Since the genus *Sarsamphiascus* has undergone many systematic changes, and because morphological differences between species are miniscule, further molecular studies are necessary to distinguish the evolutionary position of the new species within *Sarsamphiascus*.

## ACKNOWLEDGEMENTS

We would like to thank the Marine Zooplankton Resources Bank of Korea for providing specimens of *S. kawamurai* and Raehyuk Jeong for reading and revising the manuscript. We express our sincere gratitude to editor Prof. James Reimer, reviewer Prof. Viatcheslav Ivanenko, and two anonymous reviewers for their constructive comments and advice on the earlier version of the manuscript.

### Funding

This research was supported by Basic Science Research Program through the National Research Foundation of Korea (NRF) funded by the Ministry of Education (2018R1D1A1B07050117), Marine Biotechnology Program of the Korea Institute of Marine Science and Technology Promotion (KIMST) funded by the Ministry of Oceans and Fisheries (MOF) (No. 20170431) and the BK21 Plus Program (EcoBio Fusion Research Team, 22A20130012352) funded by the Ministry of Education (MOE, Korea). The funders had no role in study design, data collection and analysis, decision to publish, or preparation of the manuscript.

### Grant Disclosures

The following grant information was disclosed by the authors:
Ministry of Education: 2018R1D1A1B07050117.
Ministry of Oceans and Fisheries (MOF): 20170431.
Ministry of Education (MOE, Korea): 22A20130012352.

### Competing Interests

The authors declare there are no competing interests.

### Author Contributions

- Jisu Yeom conceived and designed the experiments, performed the experiments, analyzed the data, prepared figures and/or tables, authored or reviewed drafts of the paper, and approved the final draft.
- Wonchoel Lee conceived and designed the experiments, analyzed the data, authored or reviewed drafts of the paper, and approved the final draft.

### DNA Deposition

The following information was supplied regarding the deposition of DNA sequences:
The sequences of Sarsamphiascus are available at GenBank: 18S - MN496455, MN496456, MN541391–MN541394 –and mtCOI - MN507530; MN542379, MN542380.

### Data Availability

All specimens are deposited in National Marine Biodiversity Institute of Korea.
—Holotype 1♀ (MABIKCR00246491) Allotype 1♂ (MABIKCR00246492). Paratype 2♀ ♀ 1♂ on three slides (MABIKCR00246493–MABIKCR00246495), 2♀ ♀ dissected on 14 slides (MABIKCR00246496, MABIKCR00246497), and 2♂ ♂ dissected on six slides (MABIKCR00246498, MABIKCR00246499). 3♀ ♀ 4♂ ♂ and 1♂ on two SEM stubs, respectively (MABIKCR00246500, MABIKCR00246501).

### New Species Registration

The following information was supplied regarding the registration of a newly described species:

Publication LSID: urn:lsid:zoobank.org:pub:B93346F0-3942-4CC9-A6ED-D437D899239F
Sarsamphiascus hawaiiensis sp. nov. LSID: urn:lsid:zoobank.org:act:BACD08DD-8F16-4106-8FCE-F4C1ED96A8AC.

## Supplemental Information

Supplemental information for this article can be found online at http://dx.doi.org/10.7717/peerj.8506#supplemental-information.

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
