# Peer review of "A new species of the genus Sarsamphiascus Huys, 2009 (Copepoda: Harpacticoida: Miraciidae) from a sublittoral zone of Hawaii"

_PeerJ, doi:10.7717/peerj.8506_

## Round 0.1 · original submission · Minor Revisions

I have heard back from three expert reviewers, all of whom recommend publication but have some comments. While you may respond to these comments as you wish, I will add a couple of my own comments on top of their constructive comments.

1. I agree with the first reviewer in that the molecular phylogenetic analyses and trees need more work. You could combine trees into one figure, and make the trees much easier on the eyes, with scientific names in italics at least (this is a taxonomic paper after all). Please look at figures in other papers to gain inspiration. As well, I would like to know how your molecular evolution models were chosen. Please give you new GenBank numbers in text (now only in Table 3 I believe).

2. Unlike reviewer 2, I do not agree with removing your phylogenetic analyses of the family, and do not consider this theme to be completely independent of your species creation. Still, it may be good to more strongly integrate these two themes within your paper; at the least, your paper has given this "split" impression to reviewer 2 (although reviewer 3 quite likes the inclusion of the phylogeny). I leave the final decision as to how to deal with this comment to your discretion. As well, I believe the inclusion of even a preliminary phylogeny will be of interest to readers.
Please contact me should you have any questions, and I look forward to reading a revised version of your work.

·

Basic reporting

- The second paragraph of the introduction is difficult to read.
- The description of the molecular methods can be more compact.
- All types collected are depositing in Korea not USA which is unusual for me.

Experimental design

- Super high quality of the morphological parts.
- The molecular parts probably need some polish.

Validity of the findings

- The neighbor joining tree (Fig. 7) is not necessary to my mind.
- Fig. 8 and Fig. 9 is probably better to combine.
- I was confused by not finding a tree or a distance matrix for COI as 18S is not so informative for seeing difference between species.

Additional comments

- Some introduction words about undescribed diversity of these copepods as well as what is known about their patterns of distribution can help reader to understand importance of the work.

Reviewer 2 ·

Basic reporting

no comment

Experimental design

no comment

Validity of the findings

no comment

Additional comments

Phylogenetic analyses of Miraciidae and a new species creation have values to be published in an international journal. However, there are the following problems to be revised before acceptance.

1. It is a rule for original research papers that an article has only one theme. The MS includes two different themes, i.e. (1) new species creation with a key to copepod species of a genus, and (2) phylogenetic analysis within a copepod family. These themes are independent. Because the theme (2) is considered to have higher impact to science than the theme (1), this MS should focus the theme (2). The theme (1) should be published as a different paper.

2. The morphological description is rather redundant. Morphologies common to the genus and family, such as “rami three-segmented” in the descriptions of P1-4, should be omitted, and the diagnostic description should be limited to species-specific characters.

3. Figures should be arranged according to the order cited in the text.

4. Literature-based morphological comparison should never be concluded based on figures, except for morphologies that are easily recognizable.

5. There are problems in terminology. These are noted in the specific comments.


Specific:
Line 111. The URL cannot be opened.

Line 159. “Cephalothorax” should read “Cephalosome”. The former means the 1st prosomite fused cephalosome and 1st pediger.

Lines 160-161. Descriptions should be presented from lateral to medial parts. Since seta is a singular noun, “one” of “one seta” should be deleted throughout the MS.

Line 161. Which part of seminal receptacle in Fig. 5B is “kidney-shaped”? Please show it by arrow in Fig. 5B.

Line 161. Fig. 1D, instead of Fig. 6A, should be cited here, because spinules rows are not seen in Fig. 6A.

Line 166. Seta I is unclear in Figs. 1D and 6E, F. Please add the setal numbers in Fig. 1D.

Line 174. Such the abbreviation “seg-1” is not defined and does not appear in Huys et al. (1996).

Line 176. Only 2 pinnate setae are observed on segment 2 in Fig. 2C. Where is the 3rd seta?

Lines 181-182. Delete “on basis, setal formula. 1.0.130”, because the number of setae is explained in the next sentence. Besides, the setal formula is wrong. (The correct formula is 1.0.4.)

Lines 186-187. I could not understand the one bare seta on exp-1 in Fig. 2E. It is recommended to erase the lines expressing shell thickness, by which the bare seta can be clear with a broken line.

Lines 192-193. Fig. 2G of maxilla is difficult to distinguish each endite and seta. Besides, the trapeziform allobasis without pointed spine-like tip seems unique (doubtful). Please reexamine the maxilla of the other specimen.

Line 195. The spinule number of the longitudinal spinular row appears 21 (not 20) in Fig. 2H. Such the spinule number is generally variable and therefore the definite number could not be presented except for a description on a single specimen. The spinular row appear not on the inner margin but on the medial surface.

Line 198. There are several spinular rows on the coxa. Why was only the row at outer corner described? This one is common to the genus.

Line 199. Use the same expression for “uni-pinnate”/“unipinnate” and “bi-pinnate”/”bipinnate” throughout the MS.

Line 199. The expression “fine spinules”, which is frequently used in the text, should read “setules” throughput the MS. Spinule means small “rigid” spiniform integumental element inserting a hole which is not passing through the integument, and the same but flexible slender one is called “setule”. By using “setules”, you can use simply “spinules” instead of “coarse spinules”. Delete “inner” because of the same word at Line 198.

Line 202. The phrase “with three outer spines and one long geniculate seta and one relatively short” is geniculate and unipinnate setae distally” is confusable. According to Fig.3A Consider “with three spines, mid one of them bare and others unipinnate, laterally and two unipinnate geniculate setae apically.”

Lines 198-208. The separate is too long. Separate it into three sentences, i.e., protopod, exopod, and endopod, to make it more readable. This is also the case for P2 – P4.

Line 210. What is “hyaline frills on inner distal margin”? It is not seen on exp-1 in Fig. 3B. This is also the case for P3 (Line 221).

Line 222-225. The expression is difficult to understand. Ornamentations of spines/setae should be deleted throughout the MS except for species-specific ones, because their ornamentation pattern is well illustrated in Fig. 4A and the setal formula is in Table 2. A necessary and sufficient description without repetition of the information is smartest and generally understandable.

Line 229-230. Inner setae on exp-1 and -2 in Fig. 4B look plumose (not pinnate).

Line 234. “endopod longer than exp-2” should read “endopod longer than exp-1 and exp-2 combined”.

L240. The three inner spines on the endopodal lobe look normal and not to have “forked tip” in Fig. 5A.

Line 242. The expression “one short bare seta with spinules along ...” is awkward, because bare seta does not have spinules.

Line 250. Inner projection on P1 basis in Fig. 3C is not “claw-like” but looks a naked spine. The figure should be re-drawn from a different angle.

Line 252. This sentence has two errors. It means that enp-2 has 6 spiniform setae. Locations of two spiniform setae are not “outer distal” but at mid-length of outer margin.

Lines 253-256. Terminology of seta/spine is wrong. Two apical elements on endopodal lobe are called “spines”, while two inner elements on exopod are called “setae” despite the same form as those on endopodal lobe.

Line 307. Since there have been no information about the L/W ratios of A1 segments of S. kawamurai except for illustrations, it is assumed that the 1.4 and 1.3 in this line were measured on these illustrations by the present authors. However, such the ratios based on illustrations are not always accurate because the segments are not always set horizontally on the slide. Therefore, the difference (1) should be removed from the characters distinguishing the two species.

Line 309. Difference (4), i.e. length of the seta of female P6, should also be removed because of the same reason. Length of illustrated seta is greatly reduced if it inclines.

Line 310. Microspinules may be overlooked under a light microscope and therefore the difference (5) is questionable.

Line 311. Illustrations of terminal caudal setae by Chang et al. (2009) and Sönmez & Karaytuğ (2014) were omitted except for the naked basal part. In the original description by Ueda & Nagai (2005), illustration of the setae was naked but they gave no information about the setal ornamentation. Since the ornamentation of setae has often been omitted in previous papers, it is unknown if the terminal caudal setae of S. kawamurai is naked. Consequently, the difference (6) is questionable.

Lines 311-312. A few spinules along distal margin of the caudal rami were also seen in illustrations by previous descriptions of S. kawamurai. Therefore, the difference (7) is questionable.

Lines 304-316. In conclusion, morphologies distinguishing the two species should be only (2) and (3). Specimen-based examination is necessary to adopt the other differences.

Line 319: According to Lang (1965), P6 outer spinule (=seta?) not naked but plumose. Therefore, the difference (3) is questionable. By the same reason as that in the comment at Line 311, the difference (4) is also questionable.

Line 320. Insert “caudal” before “terminal seta”.

Line 321. “number of seta” should read “number of apical setae”, because the setal number on A2 exp of S. undosus is also three. Anyway, this number is based only on the figure and therefore Lang (1965) may have overlooked the third thin apical seta and the difference (6) is questionable.

Lines 317-321. In conclusion, morphologies distinguishing the two species should be (1), (2) and (5).

Lines 334-335. This hypothesis cannot be understood. In general, early diverged species is closer to the ancestral species than later diverged species, which diverged from early diverged species. Accordingly, if the hypothesis is correct, the early diverged new species is closer to the ancestor than S. kawamurai and S. undosus. This means the ancestor has more elongate forms and more ornamented. If so, it is necessary to explain that the evolutionary trend is from elongate to less-elongate and from ornamented to less-ornamented. Anyway, such the phylogenetic consideration should not be made from morphologies but genetic results. This is also the case for the next paragraph (Lines 336-344).

Reviewer 3 ·

Basic reporting

The authors have prepared a well-crafted taxonomic manuscript of a new miraciid species from Hawaii, with inclusion of a taxonomic key for the genus Sarsamphsiascus. They also performed a useful genetic analysis of new sequences from the genus and existing sequences from Genbank. The English language is clear for the readers and the data is accessible already.

Experimental design

The ratios mentioned in lines 325-329 are of critical importance to distinguish the species. A sense of variation from multiple measurements in the form of standard error would give the reader some confidence on these diagnostic characters.

The analyses of DNA data is reasonable given the purpose of the analysis and the availability of public data in this taxon. Phylogenetic studies in Miraciidae are badly needed and this a nice small step towards this direction. It is difficult to make substantial comments on the trees since I (and the authors) regard these topologies as preliminary. I share their opinion on the need for additional work on the group. I am just curious why did the authors apply different nucleotide models in ML and NJ analysis.

Validity of the findings

Not clear if collection permits were obtained or were even necessary but I sincerely hope that no permits will ever be needed for meiofauna, generally. There is no mention at all about the habitat (except sandy sediment) and I would like to see as many details as possible on that. Was this just a random sample while the author was in Hawaii or part of a bigger project? Is there a description of the type habitat in another project. Any other work on Harpacticoida or other meiofauna in nearby sites?

The drawings meet the modern standards of harpacticoid taxonomy and the SEMs add a little more information.

Table 1: a minor comment. Can you indicate to the reader what the X and O represent?

Additional comments

Generally, this is a good taxonomic contribution and will be of interest on copepodologists working on these groups.

---

## Round 0.2 · accepted · Accept

Your paper has been very well revised, and I can find no further issues with the work. Thus, I am happy to move this into publication. Happy 2020 - and thank you for submitting to PeerJ.